# Drought Tolerance of Legumes: Physiology and the Role of the Microbiome

Ivan S. Petrushin *, Ilia A. Vasilev and Yulia A. Markova

Siberian Institute of Plant Physiology and Biochemistry, Siberian Branch of the Russian Academy of Sciences, Irkutsk 664033, Russia; ilvasil85@gmail.com (I.A.V.); juliam06@mail.ru (Y.A.M.)
* Correspondence: ivan.kiel@gmail.com

**Abstract:** Water scarcity and global warming make drought-tolerant plant species more in-demand than ever. The most drastic damage exerted by drought occurs during the critical growth stages of seed development and reproduction. In the course of their evolution, plants form a variety of drought-tolerance mechanisms, including recruiting beneficial microorganisms. Legumes (one of the three largest groups of higher plants) have unique features and the potential to adapt to abiotic stress. The available literature discusses the genetic (breeding) and physiological aspects of drought tolerance in legumes, neglecting the role of the microbiome. Our review aims to fill this gap: starting with the physiological mechanisms of legume drought adaptation, we describe the symbiotic relationship of the plant host with the microbial community and its role in facing drought. We consider two types of studies related to microbiomes in low-water conditions: comparisons and microbiome engineering (modulation). The first type of research includes diversity shifts and the isolation of microorganisms from the various plant niches to which they belong. The second type focuses on manipulating the plant holobiont through microbiome engineering—a promising biotech strategy to improve the yield and stress-resistance of legumes.

**Keywords:** drought stress; microbiome; adaptation; legumes; Fabaceae



## 1. Introduction

Drought is one of most important stress factors that negatively affects plant development. Climate change and population growth expand the territories affected by water shortages. Drought is the main cause of reductions in crop yield and hinders the progress of agriculture in many countries [1]. Soil aridity and the spread of deserts have become a global challenge that cannot be solved using irrigation and technical measures. Water shortages deteriorate morphological, physiological and biochemical processes such as the development of leaves and roots, oxygen absorption and photosynthesis. The most drastic damage exerted by drought takes place during the critical phases of growth: seed development and reproduction. The main symptoms of drought occurring in legumes (as for other crops) are twisting, burning, wilting and premature shedding [2].

During their evolution, plants, including legumes, developed a variety of drought-resistance mechanisms to regulate their morphological and physiological characteristics such as root and leaf structure, stomatal and photosynthetic regimes, and to accumulate water. Plants are called "drought resistant" if they can grow in medium and severe drought conditions [3]. Legumes are one of the three largest higher plant groups. Many legume species are well-known food crops that can acquire soil nitrogen via symbiotic bacteria and resist metals in soil. It should be underlined that Fabaceae members have unique features and the potential to be adapted to abiotic stress [4]. To overcome the negative effect of drought, legumes attract beneficial microbes that can form a symbiosis with the host. Experimental studies show that the composition and diversity of the microbial community varies between the plant parts (leaves, roots and seeds). Drought tolerance in legumes

can be promoted by mediating the microbial community using a variety of approaches. Although drought is a well-known type of abiotic stress, existing reviews mostly cover the genetic (breeding) and physiological aspects of drought resistance in legumes [5–7]. Even Special Issues devoted to the role of the microbiome in climate change mitigation are missing reviews of recent studies on the topic [6,8].

The available literature discusses the genetic (breeding) and physiological aspects of drought resistance in legumes, neglecting the role of the microbiome. Filling this gap, in this review we consider the mechanisms used by legumes to endure droughts and drought-induced changes in the microbiome. These adaptive mechanisms are diverse and include the synthesis of phytohormones and osmoprotectants, the recruitment of microorganisms and changes in plant metabolism. In addition to legumes' own ability to attract beneficial microbes, there are a number of approaches to modifying the microbial community to make the plant more drought-tolerant, such as microbial engineering. The aim of our review is to highlight the "best practices" in producing drought-tolerant legumes and to suggest directions for further research.

## 2. Common Drought-Adaptation Mechanisms in Plants

Drought stress leads to a high concentration of reactive oxygen species (ROS), which can be extremely harmful, especially hydroxyl radicals and singlet oxygen. ROS causes cell damage, membrane and protein degradation, and lipid oxidation DNA fragmentation; eventually, these processes lead to cell death [9]. Drought stress can also change the carbon and nitrogen biogeochemical cycle, which reduces the absorption of water and nutrients by the roots and lowers the cations' conductivity (e.g., $Ca^{2+}$, $K^+$ and $Mg^{2+}$). Legumes have diverse mechanisms to reduce the amount they are affected by drought, which is common for the majority of crops. The first response to drought stress involves receiving a membrane receptor-mediated signal. The signal is then transduced to express the appropriate genes. Mitogen-activated protein kinases (MAPK) and $Ca^{2+}$-dependent protein kinases transmit the signals to the nucleus, which activates various regulons, controlling the expression of drought-resistant genes, DREB, MYB/MYC, NAC, ABRE and WRKY.

### 2.1. Role of Phytohormones

Phytohormones are crucial substances, which activate a lot of cell pathways during stress. One of the main stress phytohormones is abscisic acid (ABA). It takes part in signal acquisition from the environment as well as in the regulation of physiological and biochemical features [10]. Despite the numerous works on crops devoted to ABA pathways, the number of studies covering legumes is limited [11,12].

Salicylic acid (SA) is another important phytohormone that regulates plant growth and abiotic stress reactions [13]. Exogenous SA promotes drought resistance in plants by regulating the protein kinases' activity, as well as the chlorophyll and rubisco concentration [14]. Melatonin is an evolutionarily conserved molecule, which is contained in most living organisms and possesses biologically essential properties. The role of melatonin in drought-resistant crops is poorly understood, but a number of studies show that the exogenous treatment of several legumes promotes drought resistance through inhibiting membrane injury [15,16]. The physiological and molecular activity of melatonin in plants shows that it is an important substance to stimulate Fabaceae plants, especially under the action of abiotic stress [17].

### 2.2. Osmoprotection System

Many osmoprotectants (sugars and sugar alcohols), such as mannitol, sorbitol, inositol, trehalose, proline, ectoine, glycine and betaine, play a key role in cells' drought resistance. They inactivate ROS and stabilize proteins and membranes [18]. To regulate the osmotic pressure in cells and protect the membrane from damage, some oligosaccharides (trehalose, raffinose, fructose and saccharose) are employed. They can also act as signaling molecules. A high content of carbohydrates in plants may evidence their drought resistance, while wa-

ter shortages increase the expression of genes related to the synthesis of carbohydrates [19]. The metabolism of amino acids during abiotic stress, in which proline plays an important role, should also be borne in mind. Polyamines are low-molecular aliphatic compounds. Among them, putrescine, spermidine and spermine are most frequently met in plants. The level of endogenous polyamine is induced by abiotic stress (including by drought), but exogenous spermidine treatment overexpresses the polyamine biosynthetic genes [20].

### 2.3. Reactive Oxygen Species (ROS)

As mentioned above, prolonged drought leads to the excessive accumulation of ROS in plant tissues. To protect plants against ROS, bacteria with antioxidant potential are used. These bacteria support biochemical changes (content of proline, proteins, and antioxidant enzymes) and promote plant growth [21]. Elevated ROS levels are also controlled by ROS efflux systems, which include both non-enzymatic antioxidants and antioxidant enzymes. Furthermore, the levels of various ROS are regulated during $N_2$ accumulation in legume roots using symbiotic bacteria [9].

## 3. The Role of the Beneficial Microbes to Face Drought

In nature, plants co-exist with various microorganisms, such as viruses, bacteria, archaea, oomycetes, and fungi. Plants usually attract various kinds of microbes to promote their growth, and all of them interact with each other in a complex manner. Some experts even call these microorganisms "second genome of the plant". The soil microbiome protects plants against drought and improves the yield and soil fertility. Many studies highlight the importance of rhizosphere microbiome in improving drought resistance. It seems obvious that attracting beneficial microbes is a common evolution strategy for plants under water shortages [22,23]. The plant microbiome has attracted the interest of the agricultural research community in terms its uses in sustainable crop production and food security [24]. Soil microbes take part in a variety of processes that are crucial for plant productivity: nutrients circulation, soil mineralization, resistance to diseases and overcoming abiotic stresses (high salinity and drought). Many legumes are known to be involved in such plant–microbe interactions [25].

Generally, a plant can be considered as a holobiont and unified biological object of evolution (Figure 1). During mutual adaptation, many plant species (including legumes) formed close symbiotic relationships with bacteria and fungi. Such symbioses allow for the acquisition of a rich spectrum of nutrients that would be unavailable without the symbionts—plants themselves lack the necessary enzymatic systems (such as nitrogen fixation). Symbiotic relationships bring advantages to both sides: microsymbionts obtain access to the host's resources, while the macrosymbiont (host) uses microbial metabolites to enrich nutrition and resist abiotic stresses. From an evolutionary perspective, the symbiose leads to the pooling of hereditary information towards more diverse, top-specie heredity systems, which accelerate the evolution processes [26]. Drought stress is a major selection factor, shaping the rhizospheric drought-resistant microbiome over many seasons. This trait may be inherited from the entire hologenome of the plant and its biota, i.e., it indicates how the plant can memorize multiple past drought stress events [27]. A prolonged drought period may irreversibly change the microbiome. Recently, Santos-Medellín et al. showed that, even after removing drought stress, the endophytic microbiome could not be regenerated to the initial state. During long periods of water shortage, the microbiome configuration changes and plant health is damaged [28]. Let us take a closer look at each of the hologenome niches.

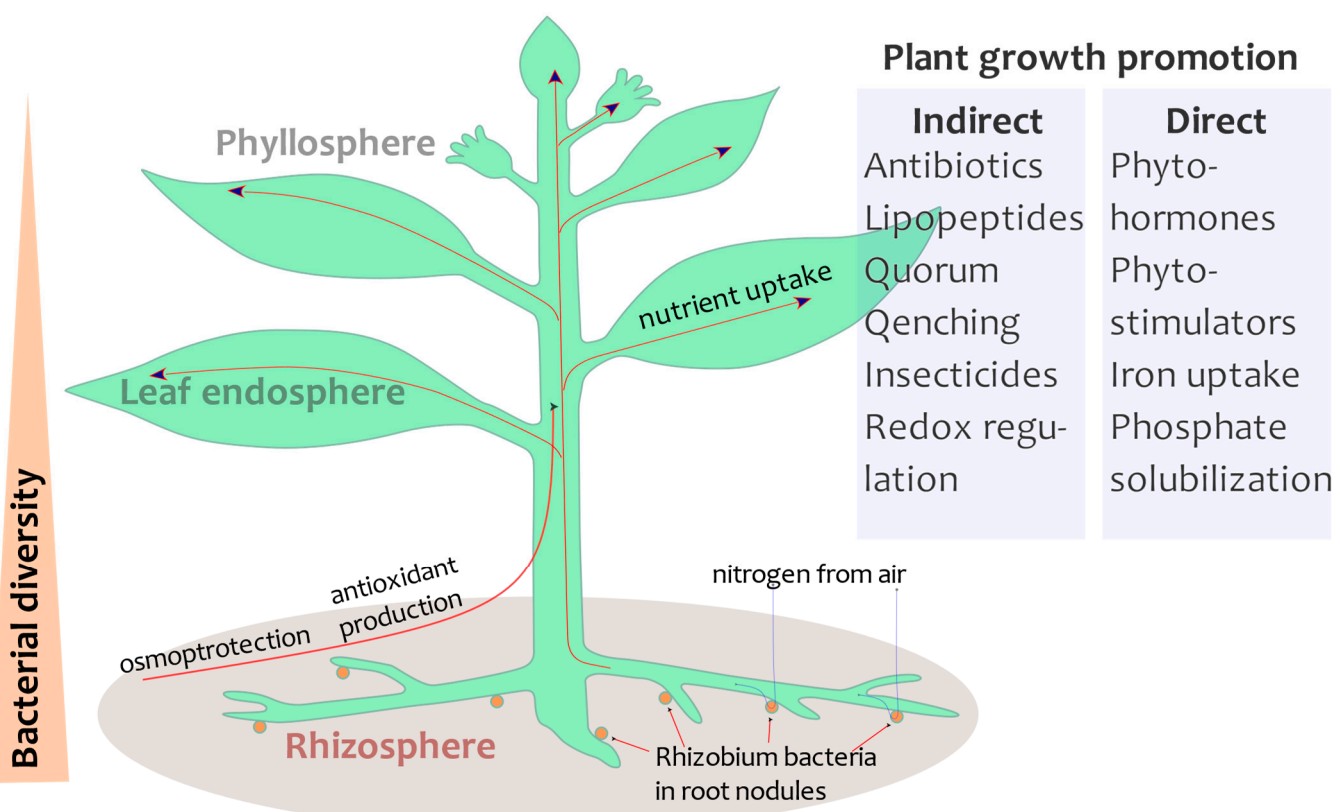

**Figure 1.** Fabaceae plant hologenome. Legumes interact with microorganisms in the soil and air. The composition of the microbiome depends on the part of the plant. Usually, three compartments, the rhizosphere (near root soil), endosphere (inner tissue of the plant) and phyllosphere (leaves and stems surface), are considered. The beneficial bacteria can promote plant growth by direct and indirect mechanisms.

A number of high-throughput sequencing studies show that the composition of the microbial community differs significantly between plant parts (leaves, roots, seeds, and rhizosphere). In general, plant host characteristics, species, age, crone type, genotype and sterility can significantly change the microbial composition. Moreover, the plant host genotype determines the profile of microbial community members, as reported for soybean [29]. Usually, the population of microbes is highest in the soil and decreased in the rhizosphere, phyllosphere and endosphere (Figure 1), showing the selection gradient [30].

Phyllosphere has a more dynamic environment than the endosphere and rhizosphere (Figure 1) because its microbiome has fewer common taxa than the endosphere and rhizosphere. To colonize various parts of a plant, microbes use a vast arsenal of tools: biofilms, biosurfactants, quorum sensing, pilis, flagella, adhesion molecules, etc. To protect plants from sunlight (UV), epiphytic bacteria synthesize the pigments. *Ascomycota* and *Acidobacteria* are dominating phyla in the phyllosphere and rhizosphere, as reported for the soybean [31]. Compared to the leaf surface, the endosphere is likely to be richer in nutrients and has a more stable environment (protected from fluctuations in the atmosphere, including UV radiation, temperature and moisture). Otherwise, endophytes have a closer interaction with the plant host immunity system that restrains bacteria reproduction.

To attract the beneficial bacteria in soil and leaves, plant uses exudates. These substances can contain amino and fatty acids, sugars, growth factors, vitamins, etc. With the help of secondary metabolites (flavonoids, coumarins, citrates oxalates), plants can recruit specific types of microbes in the rhizosphere, phyllosphere and endosphere [30,32]. Some researchers call this action a "cry for help" [33,34]. This phenomenon is well-known for the nitrogen fixation with Rhizobia and plant-growth-promoting fungi in cases where there are low levels

of nitrogen and phosphates in soil [35]. Their high ability to acquire atmospheric nitrogen is the key feature of legumes. The symbiosis of Fabaceae plants with soil bacteria (rhizobia) represents one of the best-known mutualistic plant–microbe interactions because of its contribution to the sustainability of agricultural systems and human nutrition [36]. This interaction is so close that it even affects the flowering pathways, as was shown for soybean [37]. During symbiosis, rhizobia fixate on atmospheric nitrogen, which becomes available to the plant. The amount of acquired nitrogen often meets most of the plant's needs, and the nitrogen retained in the soil becomes available for the following season's crops [38].

We should note that the interaction with the root exudates of cereals affects rhizobia activity during intercropping. In [39], the greatest effect was found for maize (compared with wheat, or barley). The involvement of nitrogen-fixing symbionts has an important advantage: it allows for the use of synthetic nitrogen fertilizers, the long-term use of which seems to lead to the predominance of less effective strains of rhizobia in the agroecosystem, to be reduced. Rhizobia are not the only root symbionts: the existence of various bacterial endophytes within nodules was reported for many legumes [40,41], and experiments involving co-inoculation with rhizobia suggest that a number of endophytes associated with nodules may stimulate growth and be safe and effective partners. Rhizobial inoculants are available and easy to use, so they have been developed and employed worldwide. However, the effect of the application of outdoor rhizobia on legume productivity varies widely and seems to depend on both environmental constraints and cultivation history. Recently, the advantages of legume selection for $N_2$ fixation in parallel with crop rotation (versus the intercropping of grain legumes) in small-scale agriculture in Africa have been highlighted [42].

Although the legumes can attract the beneficial microbes themselves, a number of approaches can mediate the microbial community to make the plant more tolerant toward drought. These approaches will be considered in the following sections.

## 4. Microbiome Engineering

The mechanisms of plant–bacterial communication are still poorly understood, but this knowledge is crucial for the further engineering of microbiomes with particular features [43,44]. This kind of host-mediated microbiome engineering was described for the soil microbiome of Arabidopsis, when plants selected microbes that would help to change the leaf biomass and flowering time [45], and may develop seeds before drought, which would cause plant disease or death. The endophytic microbiome is sensitive to drought and quickly responds to drought stress. As a result, diversity rises and shifts, while the interaction between plants and endophytes intensifies [46]. There are two main microbiome-engineering approaches to overcome drought stress: "synthetic communities" (SynComs) [47,48] and "host-mediated microbiome engineering" (HMME) [49]. Both approaches have recently been applied to legumes, but the number of studies where the authors clearly define the approach used as HMME or SynComs is limited: SynComms for *Medicago sativa* [50], and for *Crotalaria juncea*, and *Canavalia ensiformis* [51]. Nevertheless, new experiments and methods for Fabaceae species can be suggested based on the results for other crops, some of which are briefly described below.

The first approach (SynComs) deals with the design of inoculants using microbial ecology and genetics approaches, as well as functions, which could improve plant characteristics and promote plant–microbe and microbe–microbe interaction [52]. For example, Rolli et. al. developed SynCom using *Bacillus*, *Acinetobacter*, *Sphingobacterium*, *Delftia* and *Enterobacter* for grapes, which not only protect plants during drought, but promote growth and yield [53]. A similar approach was applied to blue maize when combined inoculation with several bacterial strains (*P. putida* KT2440, *A. brasilense* Sp7, *Acinetobacter* sp. EMM02, and *Sphingomonas* sp. OF178) promoted growth better than a monoculture. This bacterial consortium possesses desirable features for application in sustainable agriculture, even for different maize varieties [54]. Using synthetic biology approaches, we can construct specialized SynComs, selecting members of the community to evaluate the impact of each bacterial strain.

Host-mediated microbiome engineering is an innovative approach to developing long-term beneficial microbiome features when the host phenotype is employed for the indirect iterative selection of microbiomes. Its main advantage over the SynComs approach is in the fact that most selected microbes are adapted to stress conditions and have a strong relation to the plant host. Despite the elegant concept, this approach usually has modest efficiency and the selection process can be unsuccessful [55]. The number of studies using the HMME approach has been limited to date, but some of the results are encouraging. For example, to protect *Brachypodium distachyon* from salinity stress, Muller et al. applied the HMME and defined the beneficial microbiome. Some microbial communities increased the seed yield to 55–205% in comparison with the control (in addition to salinity resistance) [56]. An important question is how to maximize the impact of the genetic effects encoded by the microbial community to the host traits. A promising strategy is to limit or infer the host's genetic contribution to the phenotype of the whole system (plant–microbiome). The inbred or cloned plant populations could be used to minimize genetic variation. Under such conditions, genetic effects encoded by the microbial community become the major factor [57].

The typical scenario of root microbiome changes during drought was documented in a number of works. Diversity shifts towards Gram-positive bacteria, especially *Actinobacteria*, while the Gram-negative ones (the usual population of the rhizosphere) lose their niche. This Gram-positive enrichment is proportional to drought duration and severity [58]. When water returns to the soil, the microbiome quickly returns to its original state. Several hypotheses were proposed to explain this conservative pattern [59]. In general, these hypotheses are based on metatrancriptome data of microbial communities suffering from drought, but a detailed analysis is hard to perform because information about the functional and genetic features of rhizosphere community members is lacking [60]. Manipulating the plant holobiont through microbiome engineering is a promising biotechnology strategy to improve the yield and stress resistance of legumes. In the next section, we review two types of studies: inoculation of Fabaceae plants with microbes (with sole and mixtures), and isolation of microbial cultures for the further inoculation of non-legumes.

## 5. Microbiome Modulation of Fabaceae Plants

A lot of work was devoted to plant-growth-promoting rhizobacteria (PGPR), but most of it was not focused on specific plant species or families [61]. An important approach to overcome the negative impact of drought is various types of inoculation (seeds or soil treatment with microbial mixtures). Here, we briefly describe some recent successful efforts in this direction for legumes.

It seems that the *Pseudomonas* species is a very common beneficial component in drought-resistant bacterial mixture or in sole action. In many studies, these bacteria were isolated from soil roots or used as biofertilizers. *Pseudomonas* bacteria can synthesize indole acetic acid (IAA), 1-aminocyclopropane-1-carboxylate-deaminase (ACC), siderophores and successfully colonize roots by forming biofilms. The ability of *Pseudomonas* to alleviate drought stress in *Vigna radiata* was evaluated by Uzma et al. Five *Pseudomonas* were isolated and used as bioinoculants [62]. Conversely, the drought-tolerant species can be a source of specific bacterial isolates. An effort to move microbiome components (bacteria and fungi) to improve drought resistance was made using *Alhagi sparsifolia*, a known desert plant. Lei Zhang et al. extracted microbes from the plant host rhizosphere and isolated the *Pseudomonas* strain LTGT-11-2Z cell culture. When introduced to the wheat soil, this cell culture improved wheat drought resistance [63]. The isolated microbes can also help plants adapt to the non-natural environment in the case of transplantation. The composition of the microbiome is an important factor for the growth of wild plant species in the field or greenhouse. Zuo et al. tried to transplant the natural-habitat soil fungal community to the pot experiment. These fungi (species *A. chlamydospora*, *S. kiliense*, and *Monosporascus* sp.) showed high survivability under drought stress, which appeared to be developed during their long-term adaptation to low water conditions [64].

In addition to microbiome manipulation, the genetics are also studied, but such works are rare for legumes. Most studies are focused on the microbiome composition, neglecting the genetic features of particular beneficial bacteria, the expression levels of genes related to plant growth promotion and drought resistance. To provide further insight into plant–bacterial interactions under stress conditions, Nishu et al. isolated the *Pseudomonas fluorescens* DR397 and performed in vitro polyethylene glycol-based screening experiments [65]. As a result, the versatile strain *Pseudomonas fluorescens* DR397 could be used as a promising biofertilizer, improving plant drought tolerance. We previously mentioned the common beneficial factor, the expression of ACC deaminase, which reduces the concentration of ethylene in plants. Andrey Belimov and coauthors evaluated the role of the *acdS* gene using a knockout mutant. The experiment showed that the ACC deaminase of rhizosphere bacteria promoted the successful nodulation of pea (*Pisum sativum*) [66]. Further, the same team performed a pot experiment with pea line SGE and its Cd-tolerant mutant SGECd$^t$, which were cultivated under optimal and limited water conditions. They reported that water stress affected the rhizosphere microbiome far more significantly than plant genotype (in terms of alpha and beta diversity indices) [67].

The application of the microbial mixture instead of a monoculture seems to be a more promising approach due to the higher stability and versatility of the obtained community. To extend the biochemical activity of sole *Pseudomonas*, Mora et al. added *Bacillus* bacteria to an organic biofertilizer [68]. The authors note that many species among the *Bacillus* and *Pseudomonas* genera have plant-growth-promoting activity. This mixture allows for microbes to hydrolyze and transform complex organic molecules into simpler ones that are accessible for root adsorption. This nutrient biotransformation has a second positive effect, because biomolecules hold better and remain available to the plant's root system. The growth of fava beans (*Vicia faba*), with a mixture of *Rhizobium leguminosarum* (Rl) and *Pseudomonas putida* (Pp) added to the soil, improved water absorption and increased the expression of photosynthetic pigments [69]. A very similar study was performed with soya beans (*Glycine max*) and *Azotobacter chroococcum* (Az) and *Piriformospora indica* (Pi) bacterial species. It was reported that water deficiency reduced the growth and yield of soya bean, but the application of Az and Pi decreased the negative effect of water shortages, with no dependence on the irrigation regime being detected [70].

To extend the metabolic potential of rhizosphere organisms, a fungi–bacteria mixture was used in several studies (Table 1). Laranjeira et. al. mixed the prokaryotes (*Mesorhizobium* sp. UTADM31, *Burkholderia* sp. UTADB34 and *Pseudomonas* sp. UTAD11.3) and mycorrhizal fungi (*Rhizophagus irregularis*, *Funneliformis geosporum* and *Claroideoglomus claroideum*) in chickpea soil. As a result, crop yield increased by 6% compared with single inoculation, and by 24% compared to the control plants [71]. Long-term drought caused root degradation since the acquisition of water and nutrients stopped. To help plants to restore the root system, Yue et al. grew licorice with *Bacillus amyloliquefaciens* in near-root soil. This measure promoted root development and changed its structure [72].

Generally, modulated microbiome studies use different compositions of organisms to select the most productive solution. He et. al. described two septate endophytes of *A. mongholicus* that can colonize the rhizosphere of roots. The combined inoculation of this legume with various fungal and bacterial species showed that dual inoculation with *Paraboeremia putaminum* and *Trichoderma viride* had a stronger effect than inoculation with *Trichoderma viride* and *Acrocalymma vagum* [73]. Inoculating plants with beneficial bacteria and fungi could also help in well-watered environments. The planting of soybean, one of the most important legumes, often takes place in poor soils with an unfavorable water regime. Sheteiwy and coauthors co-inoculated soybean with mycorrhizal spores (inoculum was added to 5 g of trapped soil) and endophytic bacterium *Bacillus amyloliquefaciens* MN592674B (soybean seeds were soaked in the bacterial culture). Biofertilizers contributed to an obvious reduction in cell size and granularity, which may improve soybean tolerance to drought stress conditions [74]. The main source of this promising PGPR for drought-tolerant species is arid soils. Verma et al. isolated 50 bacterial strains from rhizosphere

samples of lobia (*Vigna unguiculata*) and performed a set of treatments with different strain combinations (all in pot experiments). The authors reported that treatment with No. T27 (*Pseudomonas* sp. IESDJP-V1 + *Ochrobactrum* sp. IESDJP-V5 + A. *brasilense*) led to more significant results (comparing the plant development to control samples and other treatments). They also note that these findings require further "in the field" validation [75]. Thus, the results may provide a platform for further understanding of the molecular mechanisms of bacterially mediated drought resistance in plants. Another important aspect of drought resistance is the microbiome's reaction to the drought stress of various severities. Legumes are known for their ability to survive in mid-arid environments, but how the diversity and shape of the microbial community depends on the irrigation regime remains poorly understood. In the next section, we cover the known studies on this topic.

**Table 1.** Studies devoted to promoting drought resistance by inoculation with the microbe mixture.

| Plant Host | Microbial Mixture Components | Reference |
|---|---|---|
| *Astragalus adsurgens* | *A. chlamydospora*, *S. kiliense*, and *Monosporascus* sp. | [64] |
| *Alhagi sparsifolia* | *Pseudomonas* strain LTGT-11-2Z | [63] |
| *Vicia faba* | *Rhizobium leguminosarum* (Rl) and *Pseudomonas putida* (Pp) | [69] |
| *Glycine max L.* | *Azotobacter chroococcum* (Az) and *Piriformospora indica* (Pi) | [70] |
| *Glycyrrhiza uralensis* | *Bacillus amyloliquefaciens* strain FZB42 | [72] |
| Chickpea (*Cicer arietinum L.*) | *Mesorhizobium* sp. UTADM31, *Burkholderia* sp. UTADB34 and *Pseudomonas* sp. UTAD11.3, *Funneliformis geosporum* and *Claroideoglomus claroideum* | [71] |
| *Vigna radiate* | *Pseudomonas aeruginosa*, the strains MK513745, MK513746, MK513747, MK513748, and MK513749 | [62] |
| *Lupinus albus* | *Bacillus pretiosus* SAICEU11[T], *Pseudomonas agronomica* SAICEU22[T] | [68] |
| *Glycine max L.* | *Bacillus amyloliquefaciens* MN592674B, Mycorrhizal spores (*Acaulospora laevis*, *Septoglomus deserticola*, *Rhizophagus irregularis*) | [74] |
| *Pisum sativum* and *Phaseolus vulgaris* | *Pseudomonas fluorescens* DR397 | [65] |
| *Pisum sativum* | *Rhizobium leguminosarum* bv. *viciae* 1066S | [66] |
| *Vigna unguiculata* | *Pseudomonas* sp. IESDJP-V1, *Pseudomonas* sp. IESDJP-V2, *Serratia marcescens* IESDJP-V3, *Bacillus cereus* IESDJP-V4, *Ochrobactrum* sp. IESDJP-V5, *Azospirillum brasilense* MTCC-4037, *Paenibacillus polymyxa* BHUPSB17 | [75] |

## 6. Legume Microbiome for Different Watering Regimes

It is particularly interesting to compare the microbiome changes that occur under drought stress when other environment conditions (plant site, salinity, nutrients, etc.) remain the same. Unfortunately, the design of the experiments in the considered studies varies significantly so they cannot be directly compared; in addition, some studies lack information on the microbial community composition (16S or ITS amplicon sequencing). However, we can highlight some common aspects. Most studies use two (drought stress and control) or three (well-watering, medium drought, severe drought) irrigation regimes for the soil or simulate drought stress with polyethylene glycol treatment. In addition [76], monocropping was used in all studies. Table 2 shows the common taxa present in both "drought" and "control" samples; a brief summary of each study is given below.

One of the negative factors during low water periods is an excessive concentration of ethylene and its precursor (1-aminocyclopropane-1-carboxylic acid, ACC). Some rhizobacteria can hydrolyze these harmful compounds and reduce their negative impact on the plant host (*Cicer arietinum*) [77]. Other bacteria have the same ACC hydrolyzation activity, particularly the *Bacillus* species. Andy et al. showed that *Bacillus* strains (*B. cereus* and *B. haynesii*) had deaminase activity, which is sufficient to overcome abiotic stress for two plant hosts (*Vigna mungo* and *Phaseolus vulgaris*) [78]. The formation of a microbiome during seedling is the initial stage of plant growth. Bintarti et al. focused on the endophytic community of the dormant seeds of *Phaseolus vulgaris* and its changes during drought. They hypothesize that seed microbiome likely characterizes taxa that have been transferred from parent to seed. Growing common beans in pot experiments with different irrigation regimes showed that, under stress conditions, diversity shifts were much higher in the bacterial community than in the fungal community [79].

A common source of both oil and protein, peanut (*Arachis hypogaea* L.), has important advantages: self-pollination and aerial flowering. Its main disadvantage is sensitivity to monocropping (and this sensitivity increases during planting years). We found four microbiome studies of peanut related to drought tolerance; common taxa are highlighted with bold font (Table 2). In greenhouse experiments, Dai et al. compared the diversity and composition of the peanut microbial community in control and drought conditions. Six major phyla were dominant in all samples (*Actinobacteria*, *Proteobacteria*, *Saccharibacteria*, *Chloroflexi*, *Acidobacteria*, and *Cyanobacteria*), which are common for many plant species. However, three of them, *Actinobacteria*, *Acidobacteria* and *Proteobacteria*, seem to have mutualistic relationships in peanut soil [80]. Although the PGPR has wide metabolic potential, the role of the fungal community cannot be neglected. The arbuscular mycorrhizal (AM) fungi are known to enhance plant growth in many species. Xu and coworkers compared the microbial community in natural, drought and fungal-mediated soil conditions [81]. The authors showed that peanut plant drought resistance inreases when the rhizosphere community contains AM fungi. The application of biofertilizers can be combined with the intercropping approach. Intercropping with Mulberry (*Morus alba* L.) is popular for planting peanuts in China. Li et al. reported that, in an experiment with three plant configurations, pure mulberry planting, pure peanut planting, and mulberry and peanut intercropping, there were significant differences in the bacterial and fungal communities [76]. As with most legumes, peanut plants have a close relationship with soil organisms, and in monocropping practices they become more sensitive to fungal pathogens from the soil. To compare the transcriptional response to monocropping combined with drought, Luo et al. performed both field trial and pot experiments with peanut. The authors revealed that long-term monocropping altered the soil structure, raising the percentage of small aggregates and lowering water availability. Monocropping practices increase the severity of drought stress [82].

Studies of microbiome phylogeography are of particular interest: they allow for the determination of the core species of microbiome and match the environmental factors at each plant site with the specialized bacteria. The symbiotic efficiency of fenugreek (*Trigonella foenum-graecum*) rhizobia depends on the bacterial strain and environmental conditions. Khairnar et al. performed a phylogenetic analysis of housekeeping genes, which revealed unique genotypes of fenugreek rhizobia, such as *Ensifer* (*Sinorhizobium*) *meliloti*. These strains are characteristic of agroclimatic regions of India and differ from other known genotypes [83]. Understanding the mechanisms of the mutual interaction of plant-associated microbes in different niches is key to the promotion of plant growth. Shirley Evangilene and Sivakumar Uthandi compared the diversity of the bacterial community in four niches—soil, rhizosphere, root nodules and seeds—of the horse gram (*Macrotyloma uniflorum*). They reported that the ammonium-oxidizing metabolism (*amoA*), nitrite-reducing metabolism (*nirK*) and nitrogen-fixing metabolism (*nifH*) were common and prominent in all niches, but the alpha diversity showed no significant difference. The obtained microbial cultures can substitute the synthetic fertilizers and maintain soil fertility for sustainable agricultural practices [84].

Although most attention is paid to cultured edible Fabaceae representatives, some wild types and varieties are understudied. There are known species growing in arid soils. Bambara groundnut is one of them. It can survive in marginal soils and become tolerant to drought. Ajilogba et al. reported that Bambara groundnut could selectively modulate the composition and potential functions of its microbiome during all developmental stages [85].

We should note that some studies do not contain metagenome or amplicon sequencing data to allow for reproductions of the results. However, the isolated bacterial and fungal strains could be used as a PGPR for other species [77,78]. Although the number of studies describing the above certainly is not exhaustive, it allows for us to define the future research directions.

**Table 2.** Studies of legume microbiomes under drought stress.

| Plant Host | Major Taxa | Water Regime/Soil Type | Reference |
| --- | --- | --- | --- |
| *Arachis hypogaea* | *Acaulospora, Glomus, Gigaspora* | Well-watered 45%/30%; medium drought 30%/15% [1] | [81] |
| *Cicer arietinum* | *Azotobacter chroococcum, Bacillus subtilis, Pseudomonas aeruginosa, Bacillus pumilis*[2] | Drought stress was created by adding 32.6% of polyethylene glycol (PEG 6000) | [77] |
| *Vigna mungo, Phaseolus vulgari* | *Bacillus cereus, Bacillus haynesii*[2] | In vitro drought tolerance study was conducted using PEG 6000 | [78] |
| *Phaseolus vulgaris L.* | *Pseudomonas, Bacillus, Acinetobacter, Raoultella, Escherichia-Shigella* | Ample water (300 mL/day); 66% less water (100 mL/day); Hoagland solution (300 mL/day) | [79] |
| *Arachis hypogaea L.* | *Actinobacteria, Proteobacteria, Saccharibacteria, Chloroflexi, Acidobacteria,* and *Cyanobacteria* | 85% of field capacity (control); 45%–drought | [80] |
| *Arachis hypogaea L.* | *Leptospaerulina, Cladosporium, Apiotrichum; Actinobacteria, Proteobacteria, Acidobacteria, Chloroflexi;* | Field and pot experiment | [82] |
| *Arachis hypogaea L.* | *Ascomycota, Basidiomycota, Mortierellomycota* | Natural soil | [76] |
| *Trigonella foenum-graecum* | *Ensifer meliloti*[2] | No data | [83] |
| *Macrotyloma uniflorum* | *Proteobacteria, Actinobacteria, Firmicutes, Acidobacteria, Bacteroidetes, Planctomycetes, Gemmatimonadetes* | Bulk soil, rhizosphere soil, root nodules and seed samples | [84] |
| *Vigna subterranea* | *Actinobacteria, Proteobacteria, Acidobacteria* | Bulk soil | [85] |

[1] for the seedling and flowering stages, respectively. [2] no amplicon sequencing for microbiome analysis was performed in this study.

## 7. Conclusions

Several approaches can be used to improve drought tolerance in legumes: selection (breeding), genotype modification, agronomic methods and microbiome modulation. Although the reviewed studies are methodologically quite different, we can conclude that a single approach is not sufficient to obtain a stable and productive Fabaceae crop under drought conditions. In works on microbiome modulation, it has been shown that mixtures of bacteria and fungi are more promising biofertilizers than monocultures.

In order to achieve better reproducibility and an easy comparison of results, it seems important to develop a protocol for research on the microbiome of drought-tolerant plant species, particularly legumes. In this protocol, the irrigation regime, sample preparation and

DNA extraction technique could be standardized. Unfortunately, many studies do not pay attention to the physical and chemical properties of the soil, which undoubtedly influences the composition of the microbial community. To demonstrate the high potential of the plant microbiome in agriculture, future studies will use a complex approach combining the methods of microbiology, metagenomics, metatranscriptomics and metabolomics.

**Author Contributions:** Conceptualization, I.S.P. and Y.A.M.; writing—original draft preparation, I.S.P. and Y.A.M.; writing—review and editing, I.S.P., I.A.V. and Y.A.M.; visualization, I.S.P.; supervision, Y.A.M.; project administration, Y.A.M.; funding acquisition, Y.A.M. All authors have read and agreed to the published version of the manuscript.

**Funding:** This work was supported by the Russian Science Foundation (RSF; project no. 23-26-00204).

**Institutional Review Board Statement:** Not applicable.

**Informed Consent Statement:** Not applicable.

**Data Availability Statement:** No new data were created or analyzed in this study. Data sharing is not applicable to this article.

**Acknowledgments:** The bioinformatics data analysis was performed, in part, using the equipment of the Bioinformatics Shared Access Center, the Federal Research Center Institute of Cytology and Genetics of Siberian Branch of the Russian Academy of Sciences (ICG SB RAS).

**Conflicts of Interest:** The authors declare no conflict of interest.

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
