# Peer review of "Drought Tolerance of Legumes: Physiology and the Role of the Microbiome"

_cimb, doi:10.3390/cimb45080398_

Round 1

Reviewer 1 Report

Comments and Suggestions for Authors

The article cimb-2378374 is slightly suited to the CIMB-MDPI as per aims and scope. However, to improve the quality of the manuscript, the Reviewer has some major comments.

- Comment 1: The information on legumes is lacking in the Introduction part, especially the impacts of drought on legume production. The reviewer also did not see any information related to the phrase “microbiome” in the Introduction. Please, improve this part.

- Comment 2: Section 2 is really a misunderstanding. The author should provide a comprehensive understanding of the drought adaptive mechanisms of plants, then give a paragraph on how legume crop species adapt to drought stress.

- Comment 3: The Conclusion part must be rewritten. The author should focus on two keywords, like “legumes” and “microbiome” to conclude the whole manuscript. Also, no reference should be cited in this part, it is quite strange!

- Comment 4: The figure is poor-described. If possible, please improve it and provide another figure to describe the mechanism of how microorganism strains could affect the drought adaptation mechanisms of legumes.

- Comment 5: The whole manuscript is not really well-prepared, like the affiliation, abbreviations, italicized words, style of references, etc. Please, improve it in the revised versions.

Comments on the Quality of English Language

- Comment 6: Many standard errors were easily found in the whole text, like some Russian symbols, lacking full words in the first occurrence of abbreviation, and scientific names of bacteria strains. The author should carefully check all mistakes in the manuscript.

Author Response

Dear Editor and Reviewers!

We thank you for your generous comments on the manuscript and have revised the manuscript according to your suggestions. In this version, all comments were addressed point-by-point and revisions were made. We hope our revisions are satisfactory to you and the reviewers.

The initial manuscript was completely revised to address the topic of legumes drought tolerance. Our project is devoted to endemic Fabaceae species of Baikal region (https://rscf.ru/en/project/23-26-00204/), which grow in drought cognitions with high solar exposure. Sections 2 and 3 were too common in first version, now most of references are related to Fabaceae species, all physiological aspects (section 2) fit for legumes.

Reviewer 1

Q: - 1: The information on legumes is lacking in the Introduction part, especially the impacts of drought on legume production. The reviewer also did not see any information related to the phrase “microbiome” in the Introduction. Please, improve this part.

A: We extended the Intro section to make the accent on legumes and its microbiome. Some references were also changed according to the topic.

Q: - 2: Section 2 is really a misunderstanding. The author should provide a comprehensive understanding of the drought adaptive mechanisms of plants, then give a paragraph on how legume crop species adapt to drought stress.

A: Sections 2 and 3 were too common in first version, now most of references are related to Fabaceae species, all physiological aspects (section 2) fit for legumes. Section 2 is revised according to drought adaptation of legumes.

Q: - 3: The Conclusion part must be rewritten. The author should focus on two keywords, like “legumes” and “microbiome” to conclude the whole manuscript. Also, no reference should be cited in this part, it is quite strange!

A: The Conclusion section is completely rewritten. We summarized our thoughts and provided some suggestions.

Q: - 4: The figure is poor-described. If possible, please improve it and provide another figure to describe the mechanism of how microorganism strains could affect the drought adaptation mechanisms of legumes.

A: The figure is extended with drought adaptation mechanisms specific to legumes. Although some of them are common to other higher plants.

Q: - 5: The whole manuscript is not really well-prepared, like the affiliation, abbreviations, italicized words, style of references, etc. Please, improve it in the revised versions.

A: We tried our best to improve the text style and formatting. References were processed by Mendeley using MDPI template.

Q: The overall introduction must be improved. The purpose of this review needs to be clarified. What is the strength and uniqueness of this review? What is the current state of knowledge of drought in legumes? What topics will it cover that have yet to be covered previously? The topic of drought is well established, so the authors should focus on recent advances and narrow it down to legumes only.

A: We extended the Intro section to make the accent on legumes and its microbiome. Some references were also changed to better fit to the topic. While collecting the studies for the review, we narrowed our search to recent studies (2019 or later). As told in the Intro: “The available literature discusses the genetic (breeding) and physiological aspects of drought tolerance in legumes, neglecting the role of the microbiome.” The aim of the review is filling the gap in our knowledge about role of plant microbiome to drought tolerance.

Reviewer 2 Report

Comments and Suggestions for Authors

The review manuscript attempts to highlight the current state of knowledge on drought stress responses in legumes and the associated role of the microbiome. However, there several issues that must be address by the authors.

Title

The content does not reflect the title. Initially, the physiological topic was presented in general, not focusing on legumes. The drought effects on legumes were only discussed in Sections 5 and 6. It would be better if the earlier topics were discussed related to the latest findings in legumes to provide a comprehensive foundation. 

Introduction

The overall introduction must be improved. The purpose of this review needs to be clarified. What is the strength and uniqueness of this review? What is the current state of knowledge of drought in legumes? What topics will it cover that have yet to be covered previously? The topic of drought is well established, so the authors should focus on recent advances and narrow it down to legumes only. 

Section 5

- the authors mentioned SynComm and HMME as the two main microbiome engineering approaches in Section 4. Are the two approaches used in the microbiome engineering of legumes? It should have been covered in Section 5. 

- Can Table 1 be modified to include the effects of the microbial mixture on the specific plant host? For example, the reduction of drought effects or the increment of plant yield in %

Section 6

- highlight the common or unique microbes associated with drought stress in legumes in general and in specific plant species 

- what is the purpose of Table 2? What is the significance of including information on the 'plant site'? Is it possible to include the unique microbes identified in each study?

- the authors must present a critical discussion for this section, not just a descriptive summary of the topic

Conclusion

The section is also very general and not a sentence mentioning legumes. The authors should focus on communicating what they understand about drought, the role of microbiomes in legumes, and what remains unknown. Authors should also provide suggestions for future research. 

Comments on the Quality of English Language

Overall, the English language of the manuscript requires proofreading by a professional proofreader. 

Author Response

Dear Editor and Reviewers!

We thank you for your generous comments on the manuscript and have revised the manuscript according to your suggestions. In this version, all comments were addressed point-by-point and revisions were made. We hope our revisions are satisfactory to you and the reviewers.

The initial manuscript was completely revised to address the topic of legumes drought tolerance. Our project is devoted to endemic Fabaceae species of Baikal region (https://rscf.ru/en/project/23-26-00204/), which grow in drought cognitions with high solar exposure. Sections 2 and 3 were too common in first version, now most of references are related to Fabaceae species, all physiological aspects (section 2) fit for legumes.

Reviewer 2

Q: Section 5

- the authors mentioned SynComm and HMME as the two main microbiome engineering approaches in Section 4. Are the two approaches used in the microbiome engineering of legumes? It should have been covered in Section 5.

A: Yes, these approaches are applicable to legumes, but the number of such studies is very small. We have expanded the section 4 with the examples. We have also changed the title of the section to a broader one: “Microbiome modulation of Fabaceae plants”.

Q: - Can Table 1 be modified to include the effects of the microbial mixture on the specific plant host? For example, the reduction of drought effects or the increment of plant yield in %

A: Thank you for this suggestion. We have tried to do this before writing the manuscript, but design of the studies is very different and cannot be reduced to the common pattern for the table.

Q: Section 6

- highlight the common or unique microbes associated with drought stress in legumes in general and in specific plant species

A: We described the common taxa in the Table 2. Of course, this approach is limited due to changes in the microbiome composition in different irrigation regimes. To define the unique microbes (specific for plant species) we have to narrow the review to certain species.

Q: - what is the purpose of Table 2? What is the significance of including information on the 'plant site'? Is it possible to include the unique microbes identified in each study?

A: In the Table 2 we summarized the experimental conditions and species of each study. At your suggestion this table has been extended to include major microbial taxa.

Q: - the authors must present a critical discussion for this section 6, not just a descriptive summary of the topic

A: We agree with the Reviewer, but differences in experimental design complicate the comparison. We added the intro paragraph to the section 6 and revised the structure of the main text. At the end of the section some concluding remarks are added.

Q: Conclusion

The section is also very general and not a sentence mentioning legumes. The authors should focus on communicating what they understand about drought, the role of microbiomes in legumes, and what remains unknown. Authors should also provide suggestions for future research.

A: The Conclusion section is completely rewritten. We summarized our thoughts and provided some suggestions.

Reviewer 3 Report

Comments and Suggestions for Authors

The manuscript entitled “Drought resistance of legumes: physiology and role of the microbiome” is a review article that focuses on the role of the microbiome in the drought resistance of legumes, specifically the Fabaceae family. The topic of drought resistance is of great importance, especially in the context of global warming and the need for sustainable agriculture. The manuscript highlights the potential of legumes as a crop with unique features that could be adapted to abiotic stress, which could be useful in developing drought-resistant crops. Overall, the manuscript is informative and relevant to the current scientific research in the field of plant-microbe interactions and drought adaptation.

However, in my opinion, the quality of the manuscript needs to be considerably improved before it can be published, as there are several issues that need to be addressed:

1. While the abstract provides a brief overview of the content covered in the manuscript, the sentences do seem to lack a clear flow and connection between them. The abstract would benefit from some restructuring to improve its coherence and readability to make it more informative and engaging for readers.

2. Secondly, the use of "i.e." and "f.e." is inconsistent, as the authors use "i.e." and "f.e." interchangeably throughout the manuscript, which could potentially lead to confusion for the reader.  "i.e." stands for "id est," which is Latin for "that is," whereas "f.e." is just a common English abbreviation and is not typically used in academic writing. I suggest the authors use "e.g." which stands for "exempli gratia," which is also Latin and means "for example." instead of "f.e. "

3. Found some Cyrillic letters throughout the manuscript.

4. The authors overuse the definite article "the", while some sentences lack the proper use of articles. Therefore, it is recommended to revise the manuscript for consistency in the use of articles.

5. Several grammatical errors can be found in the manuscript. As an example, the very first sentence in the abstract "The water deficit and global warming makes drought-resistant plant species needed more than ever" should be corrected to "The water deficit and global warming make drought-resistant plant species needed more than ever."

6. There is a lack of connection between some sections of the manuscript, which makes it difficult for the reader to follow the flow of ideas. Therefore, the authors are advised to ensure that each section connects logically to the next. some loose phrases throughout the document that need better contextualization.

7. In addition to the issues previously discussed, the manuscript also suffers from an overuse of short, disconnected sentences. This further contributes to the lack of continuity and cohesion in the writing, making it difficult for the reader to follow the author's line of thought. It is important to combine and connect ideas to create a coherent narrative that is easy to follow.

I hope that you find these comments helpful.

Comments on the Quality of English Language

There are several grammatical errors, awkward phrasings and other language issues throughout the manuscript, that make it difficult to understand and detract from the overall quality, indicating that the quality of English could be considerably improved. Assuming that English may not be the authors' first language, probably the manuscript could benefit from the assistance of a native English speaker or a professional editor to help improve the clarity and flow of the writing.

Author Response

Dear Editor and Reviewers!

We thank you for your generous comments on the manuscript and have revised the manuscript according to your suggestions. In this version, all comments were addressed point-by-point and revisions were made. We hope our revisions are satisfactory to you and the reviewers.

The initial manuscript was completely revised to address the topic of legumes drought tolerance. Our project is devoted to endemic Fabaceae species of Baikal region (https://rscf.ru/en/project/23-26-00204/), which grow in drought cognitions with high solar exposure. Sections 2 and 3 were too common in first version, now most of references are related to Fabaceae species, all physiological aspects (section 2) fit for legumes.

Reviewer 3

Q: 1. While the abstract provides a brief overview of the content covered in the manuscript, the sentences do seem to lack a clear flow and connection between them. The abstract would benefit from some restructuring to improve its coherence and readability to make it more informative and engaging for readers.

A: We have revised the abstract completely to make the accent on legumes, microbiome studies and aim of our review.

Q: 2. Secondly, the use of "i.e." and "f.e." is inconsistent, as the authors use "i.e." and "f.e." interchangeably throughout the manuscript, which could potentially lead to confusion for the reader.  "i.e." stands for "id est," which is Latin for "that is," whereas "f.e." is just a common English abbreviation and is not typically used in academic writing. I suggest the authors use "e.g." which stands for "exempli gratia," which is also Latin and means "for example." instead of "f.e."

  1. Found some Cyrillic letters throughout the manuscript.

A: Thank you for your patience, all these issues are fixed in manuscript.

Q: 4. The authors overuse the definite article "the", while some sentences lack the proper use of articles. Therefore, it is recommended to revise the manuscript for consistency in the use of articles.

A: We asked for proofreading and changed some sentences.

Q: 5. Several grammatical errors can be found in the manuscript. As an example, the very first sentence in the abstract "The water deficit and global warming makes drought-resistant plant species needed more than ever" should be corrected to "The water deficit and global warming make drought-resistant plant species needed more than ever."

A: We have revised the abstract completely.

  1. There is a lack of connection between some sections of the manuscript, which makes it difficult for the reader to follow the flow of ideas. Therefore, the authors are advised to ensure that each section connects logically to the next. some loose phrases throughout the document that need better contextualization.

A: We added the paragraphs which connect the sections together. The structure of text into sections is revised too.

Q: 7. In addition to the issues previously discussed, the manuscript also suffers from an overuse of short, disconnected sentences. This further contributes to the lack of continuity and cohesion in the writing, making it difficult for the reader to follow the author's line of thought. It is important to combine and connect ideas to create a coherent narrative that is easy to follow.

A: The structure of text into sections is revised to more easier reading. Unfortunately, the design and results of covered studies are very different and cannot be reduced to common pattern.

Round 2

Reviewer 2 Report

Comments and Suggestions for Authors

The authors have responded to all the comments, and the current version of the manuscript has improved a lot. 

Comments on the Quality of English Language

Only minor spell check is needed. 

Reviewer 3 Report

Comments and Suggestions for Authors

The authors have diligently addressed the reviewer's comments and suggestions. The authors carefully considered all the feedback and incorporated necessary corrections throughout the manuscript. Additionally, the language and writing style have been refined, making the paper more coherent and reader-friendly.

As a result of these thorough revisions, the manuscript is now significantly clearer and more compelling.

Based on the substantial improvements made by the authors, I am confident that the manuscript is suitable for publication.